# Differential Plasma Metabolites between High- and Low-Grade Meningioma Cases

**DOI:** 10.3390/ijms24010394

**Published:** 2022-12-26

**Authors:** Gabriel A. Kurokawa, Pedro T. Hamamoto Filho, Jeany Delafiori, Aline F. Galvani, Arthur N. de Oliveira, Flávia L. Dias-Audibert, Rodrigo R. Catharino, Maria Inês M. C. Pardini, Marco A. Zanini, Estela de O. Lima, Adriana C. Ferrasi

**Affiliations:** 1Laboratory of Molecular Analysis and Neuro-oncology, Department of Internal Medicine, Botucatu Medical School, São Paulo State University, Botucatu 18618-970, Brazil; 2Department of Neurology, Psychology and Psychiatry, Botucatu Medical School, São Paulo State University, Botucatu 18618-970, Brazil; 3Innovare Biomarkers Laboratory, School of Pharmaceutical Sciences, University of Campinas, Campinas 13083-877, Brazil

**Keywords:** biomarkers, grading, meningioma, metabolomics, plasma samples, screening

## Abstract

Meningiomas (MGMs) are currently classified into grades I, II, and III. High-grade tumors are correlated with decreased survival rates and increased recurrence rates. The current grading classification is based on histological criteria and determined only after surgical tumor sampling. This study aimed to identify plasma metabolic alterations in meningiomas of different grades, which would aid surgeons in predefining the ideal surgical strategy. Plasma samples were collected from 51 patients with meningioma and classified into low-grade (LG) (grade I; n = 43), and high-grade (HG) samples (grade II, n = 5; grade III, n = 3). An untargeted metabolomic approach was used to analyze plasma metabolites. Statistical analyses were performed to select differential biomarkers among HG and LG groups. Metabolites were identified using tandem mass spectrometry along with database verification. Five and four differential biomarkers were identified for HG and LG meningiomas, respectively. To evaluate the potential of HG MGM metabolites to differentiate between HG and LG tumors, a receiving operating characteristic curve was constructed, which revealed an area under the curve of 95.7%. This indicates that the five HG MGM metabolites represent metabolic alterations that can differentiate between LG and HG meningiomas. These metabolites may indicate tumor grade even before the appearance of histological features.

## 1. Introduction

Meningiomas (MGMs), which are the most common intracranial primary tumors, represent approximately 36% of intracranial primary tumors [1,2,3]. Women aged approximately 60–70 years are susceptible to MGMs [3,4,5] and the incidence of MGM increases with age [1,6]. Most MGMs, which originate from the arachnoid meningeal cells [6,7], exhibit the characteristics of slow-growing benign tumors [1,2,5,8]. According to the World Health Organization (WHO), MGMs are classified into grades I, II, and III. Grade I MGMs are benign tumors, account for 80–90% of the cases [2,7], and exhibit slow growth [1,5] with occasional mitoses and pleomorphic nuclei [4]. Grade II or atypical MGMs account for 5–7% of all MGMs and exhibit increased mitotic activity. Additionally, grade II MGMs exhibit at least three of the following features: small cells with a high nucleus/cytoplasm ratio; prominent nucleoli; increased cellularity; necrotic foci; and uninterrupted, patternless, or sheet-like growth. However, the criterion for classifying MGM as a grade II tumor is brain invasion, even in the absence of these features. In contrast to that of grade I MGM, the recurrence rate of atypical meningioma is 29–52%. Grade III or anaplastic MGMs exhibit significantly increased mitotic activity, high brain invasion rate, extensive necrosis, and malignancy features that are more severe than those in atypical MGMs. Although grade III MGMs account for 1–3% of MGMs, the prognosis is poor with recurrence rates of 50–94% [2,3,4,5].

Currently, the initial diagnosis of MGM is based on imaging techniques, such as magnetic resonance imaging (MRI) and computed tomography (CT) [2,5,8]. Additionally, tumor grade and prognosis are evaluated using histopathological analysis [9]. However, these approaches are not recommended for all patients. In the cases of patients with pacemakers, or asymptomatic MGMs, a “wait and see” approach is commonly used, and a biopsy is not performed [3]. The absence of imaging follow-up and/or histological analysis hampers tumor classification, negatively impacting diagnosis and prognosis. Although MRI, CT, and histology are useful to classify MGMs, they cannot accurately define the prognosis [10]. The identification of grades is fundamental for determining the prognosis of patients, especially for those with grades II and III tumors. Hence, the development of practical, minimally invasive, and accurate techniques is critical for the healthcare system [3,11]. To improve the accuracy of MGM prognosis prediction, the correlation between molecular features, especially those of plasma, can potentially aid MGM cases, including asymptomatic and imaging-ineligible cases. Thus, the identification of distinct biomarkers for individual MGM grades can aid in the development of an integrated diagnostic protocol with improved accuracy, powerful predictive value for recurrence and outcomes, and the most effective treatment for each case. For patients who will undergo surgical resection, preoperative information on the tumor grade will aid in planning an aggressive or conservative resection.

Untargeted metabolomic studies can contribute to the identification of differential metabolites between MGM grades. Mass spectrometry (MS) is the most sensitive technique for metabolomic studies and can detect metabolites in different biological samples, such as urine, cerebrospinal fluid, and plasma/blood serum [12,13,14]. Furthermore, MS is a useful technique because it is less invasive and highly accurate in identifying the plasma metabolite biomarkers in different tumor grades. Therefore, this study aimed to identify differential plasma metabolites between low-grade (LG; grade I) and high-grade (HG; grades II and III) MGMs, and evaluate the potential of HG MGM metabolites as tumor grade biomarkers.

## 2. Results

### 2.1. Patients

Among the patients enrolled in the study, 33 (64.7%) and 18 (35.3%) were women and men, respectively. The median and mean ages of participants were 53 (range: 27–85 years) and 53.1 years, respectively. The mean ages of patients with LG and HG tumors were 54.6 and 44.8 years, respectively.

### 2.2. Metabolites

Partial least squares-discriminant analysis (PLS-DA) revealed a clear separation between LG and HG MGM cases (Figure 1). Volcano plot analysis was performed to select the significant metabolites with *m*/*z* ratios with *p* < 0.05 and fold change ≥ 1.5 (Log_2_(FC) ≥ 0.585) (Figure 2). In total, nine differential biomarkers (four for LG MGMs and five for HG MGMs) were selected. The four LG MGM biomarkers were represented by *m*/*z* nominal values of 1294, 710, 577, and 1245. For HG MGM biomarkers, the selected *m*/*z* values were 294, 908, 800, 910, and 985. A heatmap was designed to examine the behavior of these nine biomarkers depending on the tumor grades (Figure 3). The heatmap revealed that the frequency of five HG MGM biomarkers in patients with HG MGM was higher than that in patients with LG MGM, indicating that these five HG MGM metabolites might serve as worse progression markers. To evaluate the potential of these five HG MGM *m*/*z* values as differential high-grade MGM metabolites, a receiving operating characteristic (ROC) curve was constructed considering a 95% confidence interval. The area under the curve (AUC) corresponded to 0.958 (Figure 4). In addition to *m*/*z* selection from both groups, this study identified the biomarkers. From the nine selected *m*/*z* values, the following seven biomolecules were identified (detailed in Table 1): tyrosine (*m*/*z* 294), sulfatide (*m*/*z* 908), two phosphatidylserine (PS) molecules (*m*/*z* 800 and *m*/*z* 910), lactosylceramide (LacCer; *m*/*z* 985), phosphatidylethanolamine (PE; *m*/*z* 710), and diacylglycerol (DG; *m*/*z* 577).

## 3. Discussion

### 3.1. HG MGM Metabolites

Among the selected plasma metabolites of patients with HG MGM, a sulfatide (*m*/*z* = 908) and LacCer (*m*/*z* = 985), which are both byproducts of ceramide metabolism [15], were identified. Ceramide, which is the precursor of several sphingolipids, glycosphingolipids, and gangliosides, exhibits both structural functions in the lipid bilayer and second messenger functions in the signaling and regulation of cellular processes [15,16], such as senescence, apoptosis, differentiation, and cellular growth [17,18,19].

One of the main roles of ceramide is to activate signaling cascades that lead to cell death, functioning as a tumor suppressor [20,21]. Hence, one of the routes used by cancer cells to escape immune response is the metabolization of ceramide (Figure 5A) to inhibit apoptosis and support cell growth [18,19,20,21,22]. The main escape route used by cellular machinery to suppress this antitumor function is the glycosylation of ceramide to glucosylceramide (GlcCer) by the enzyme GlcCer synthase (GCS; also known as ceramide glucosyltransferase), which transfers glucose to ceramide [18,19,21,22,23]. This effect was first studied by Liu et al. [24] who evaluated human breast carcinoma cells with increased GCS expression. Breast cancer cells were transfected with a GCS-encoding gene. Compared with that in control cells, the survival rate was higher in transfected cells. In contrast, the inhibition of this pathway increased the ceramide-induced apoptosis rate [19] and attenuated multiple drug resistance [25]. Meanwhile, the overexpression of GCS increased the resistance of cells to antitumor drugs [26].

In addition to GlcCer production, the inhibition of apoptosis via the ceramide glycosylation pathway was proposed by Beier and Görögh [27]. The authors suggested that the defense mechanism of tumor cells can involve the activity of ceramide galactosyltransferase (UGT8), an enzyme that catalyzes the transfer of galactose to ceramide to generate galactosylceramide (GalCer) [28]. UGT8 overexpression is reported to be directly correlated with the malignancy grade of tumor cells [29,30]. Additionally, both sphingolipids (GlcCer and GalCer) were correlated with multidrug resistance in colon adenocarcinoma cells [31], indicating that ceramide metabolism is associated with tumor cell survival. As the presence of these molecules can indicate changes in ceramide glycosylation, the biochemical pathways in which they are involved are potential therapeutic targets to control tumor development.

Based on these pathways, two of the selected metabolites in this study were involved in the tumor cell escape pathway. After ceramide glycosylation, GlcCer is converted into LacCer (*m*/*z* 985) in the Golgi complex (Figure 5B) [32]. Additionally, the GlcCer levels are upregulated in multidrug-resistant cancer cells [23]. Similarly, the levels of LacCer and LacCer synthase, an enzyme that converts GlcCer into LacCer, are upregulated [33,34]. Increased LacCer levels may be associated with a worse prognosis and an increased risk of metastasis in patients with colorectal cancer [35]. In the Golgi complex, GalCer is sulfonated and converted to 3-O-sulfogalactosylceramide (*m*/*z* 908) (Figure 5B) [36]. GalCer sulfotransferase (an enzyme that catalyzes the sulfation of GalCer) was reported to be upregulated in the serum of patients with renal cell carcinoma [36] and hepatocellular carcinoma [37], in addition to the upregulation of a glycolipid sulfotransferase reported in renal carcinoma cells [38]. Moreover, cells with increased metastatic potential exhibit significantly higher expression levels of sulfotransferase than those with decreased metastatic potential [39]. These data support the hypothesis that ceramide byproducts are metabolites that can promote tumor progression and are associated with drug resistance, metastatic potential, and poor prognosis.

Another molecule identified as a potential HG MGM metabolite was arginyl-proline (PR) (*m*/*z* 294), which was selected as the most frequent metabolite in the HG MGM group. Several dipeptides function as cancer biomarkers [40,41]. However, studies evaluating the pathophysiological roles of PR are limited. Arginine and proline are among the amino acids considered essential for the growth of cancer cells [42]. Gao et al. demonstrated that the overexpression of pyrroline-5-carboxylate reductase (PYCR; involved in proline biosynthesis) is associated with poor prognosis and cancer proliferation [43]. Additionally, some tumors, such as melanoma and hepatocellular carcinoma, are auxotrophic for arginine and must obtain this amino acid from extracellular sources to survive [44]. This can explain the upregulation of arginine in the plasma samples of patients with HG tumors. To the best of our knowledge, this is the first study to report that PR dipeptide is a potential plasma marker for HG MGM. Based on the importance of PR as a differential biomarker, further studies are required to investigate the predictive value of PR and its role in different grades of MGM.

Finally, the MS/MS profile revealed two PS molecules (*m*/*z* 800 and *m*/*z* 910). PS, a phospholipid, is predominantly localized to the inner leaflet of the plasma membrane [45]. However, PS transport to the outer leaflet is related to apoptosis, activating the prophagocytic biochemical pathway and macrophage recruitment [46,47,48]. After phagocytosis, apoptotic cells with externalized PS promote the overproduction of anti-inflammatory cytokines such as TGF-β and IL-10, and downregulate proinflammatory cytokines and chemokines [49,50,51]. Although the tumor microenvironment is not characterized by apoptotic cells, the outer membrane level of PS in cancerous cells is higher than that in healthy cells. Hence, cancer cells use the apoptosis signaling pathway to inactivate the antitumor immune response [52,53]. To escape from immunological defense mechanisms, PS upregulation at the outer leaflet of the tumor cell membrane suppresses inflammation, impairs cellular immunity, and supports tumor growth. Some researchers have suggested that targeting PS (mainly using the monoclonal antibody Bavituximab) is a potential anticancer therapeutic strategy [54,55,56]. Hence, the plasma levels of PS can indicate cancer progression and are potential therapeutic targets.

In addition to identifying biomarkers, this study evaluated the differential power of the HG biomarkers for tumor progression that can significantly impact clinical and therapeutical decisions. Five HG MGM biomarkers were used to construct a ROC curve to determine their discriminant power (Figure 4). An area under the curve (AUC) of 95.8% was achieved, implying that the five metabolites identified as HG MGM biomarkers can indicate the progression from an LG to an HG tumor. Therefore, these molecules must be further evaluated as targets for MGM progression, which will enable the establishment of a less invasive, more accurate and cost-effective tumor grading system, especially for the most aggressive and lethal HG tumors.

### 3.2. LG MGM Metabolites

In addition to the HG MGM biomarkers, two metabolites were identified as potential LG MGM biomarkers. PE (*m*/*z* 710), a glycerophospholipid, was one of the biomarkers. Similar to PS, PE is asymmetrically distributed in the plasma membrane and localized predominantly at the inner leaflet [57]. A study evaluating PE expression on the cell surface revealed that among 15 cancer cell lines, 12 exhibited downregulated levels of PE on the outer membrane. Hence, PE was selected as an LG MGM biomarker as it was downregulated in HG MGM [58]. Additionally, PE can serve as a biomarker for early tumor development, as the plasma levels of PE in patients with precervical cancer (squamous intraepithelial lesions) displayed potential for distinguishing early-stage cervical cancer from cervical cancer and healthy controls [59]. In addition, the serum PE levels in patients with early-stage hepatocellular carcinoma were higher than those in patients with late-stage hepatocellular carcinoma, indicating that serum PE levels are potential biomarkers for early-stage hepatocellular carcinoma [60]. These findings suggest a potential role of PE in carcinogenesis. Thus, glycerophospholipids can serve as biomarkers for early-stage disease.

Finally, DG (*m*/*z* 577), a central molecule involved in the synthesis of lipids, especially phospholipids and triglycerides, was identified as a potential LG MGM metabolite [61,62]. During lipid synthesis, DG can be converted to phosphatidic acid, a reaction catalyzed by DG kinases (DGKs) [63]. Among the enzyme isoforms, DGKα is the most studied and is reported to be associated with the development and progression of different types of cancer [64,65,66]. Additionally, DGKα is reported to be a potential immunosuppressor [67,68]. Compared with healthy T cells, cytotoxicity and cytokine secretion were higher in T cells in which DGKα expression was inhibited. This indicates the importance of DGK in cancer immune control [69]. Furthermore, studies on silencing and/or pharmacological inhibition of DGKα revealed an apparent correlation between downregulated DGKα expression and antitumor effects, including the inhibition of cell proliferation and invasion, colony formation, and cancer cell viability and the upregulation of apoptosis rates [70,71,72]. The overexpression of DGKα supports tumor cell proliferation [70,71]. Due to the aggressive behavior of HG MGM [73], lipid synthesis must be upregulated to meet the nutrient requirements of enhanced cell proliferation. Hence, membrane synthesis through DG consumption may be an alternative way to meet the nutrient demands of the HG tumor microenvironment. These findings demonstrate that high concentrations of DG in the blood of patients with LG MGM can indicate a less aggressive tumor. As DGKα is associated with cancer progression and DG is a potential LG MGM biomarker, future studies must investigate the plasma levels of DGKα and DG and their role in cancer development.

This study has several limitations, primarily the sample size of HG MGM cases. Recent epidemiological studies on MGMs have reported that grades II and III MGMs correspond to 8.2% and 0.7% of MGM documented WHO grades, respectively [74]. The present research represents a regional study and achieved a sample size that corresponds to the expected incidence index. Although these results represent a preliminary screening for metabolites of MGM grades in blood samples, they indicate biochemical alterations that may be reflected in the plasma samples of patients. We acknowledge the need for homogeneous groups to perform statistical analysis to extrapolate and apply the results in a real-world setting. In addition to some previously reported metabolites, this study revealed some novel ones. This indicates the importance of these metabolites in the pathogenesis and progression of MGMs. Therefore, the results of this study revealed molecular indicators of metabolic imbalance, which can improve our understanding of MGM grades. Additionally, this research identified some molecules that can serve as potential therapeutic targets for MGMs.

## 4. Materials and Methods

### 4.1. Patients and Samples

This study was approved by the Ethics Committee on Research of Sao Paulo State University (Protocol 4328-2012). All participants of the study signed individual informed consent forms. Fifty-one meningioma patients from both genders were included and divided into the following two groups: LG MGM (grade I; n = 43) and HG MGM (grade II, n = 5; grade III, n = 3).

### 4.2. Sample Preparation

All the blood plasma samples were collected on the first day of hospital admission for every patient enrolled in the present study. The blood was collected in EDTA-coated tubes and centrifuged at 3000 rpm for 5 min. The plasma was stored at −80 °C until analysis. Next, the plasma samples (20 µL) were subjected to metabolite extraction and ionization following the methods of Melo et al. [75]. For chemical ionization, formic acid 0.1% (analytical grade) was added to the final solution, which was subjected to MS analysis.

### 4.3. MS Analysis

After ionization, the samples were directly injected into a mass spectrometer (ESI-LTQ-XL; Thermo Scientific, Bremen, Germany) and analyzed under the following conditions: flow rate, 10 µL/min; capillary temperature, 280 °C; spray source voltage, 5 kV; sheath gas, 2 arbitrary units; analysis mode, positive ion mode; mass range, 100–1400 *m*/*z* (mass/charge). For each biological sample, 10 analytical replicates were analyzed.

### 4.4. Statistical Analysis and Molecular Identification

To identify the biochemical differences between HG and LG groups, the MS data were subjected to statistical analysis using the online software MetaboAnalyst 5.0 [76]. Multivariate PLS-DA was performed to evaluate the differences between HG and LG groups. To identify the discriminant biomarkers for each group, univariate analysis was performed and a Volcano plot was constructed. Parameters with FC ≥ 1.5 and *p* < 0.05 were used to define the most abundant and significant *m*/*z* values for each group. The selected biomarkers were also evaluated using a heatmap with Ward’s clustering method and Euclidean distance measurement. Among the *m*/*z* values evaluated, five of the selected markers for patients with HG MGM were used to construct a ROC curve and evaluate their potential as HG tumor differential metabolites. The biological molecules corresponding to the selected *m*/*z* ratios were identified by searching specialized metabolomic databases, such as the Human Metabolome Database (HMDB), METLIN, and LIPID MAPS. Next, the structure of the molecules was verified using tandem MS analysis. After identification, the biomarkers were subjected to in silico search to identify the biochemical pathways involved in carcinogenesis and tumor progression.

## Figures and Tables

**Figure 1 ijms-24-00394-f001:**
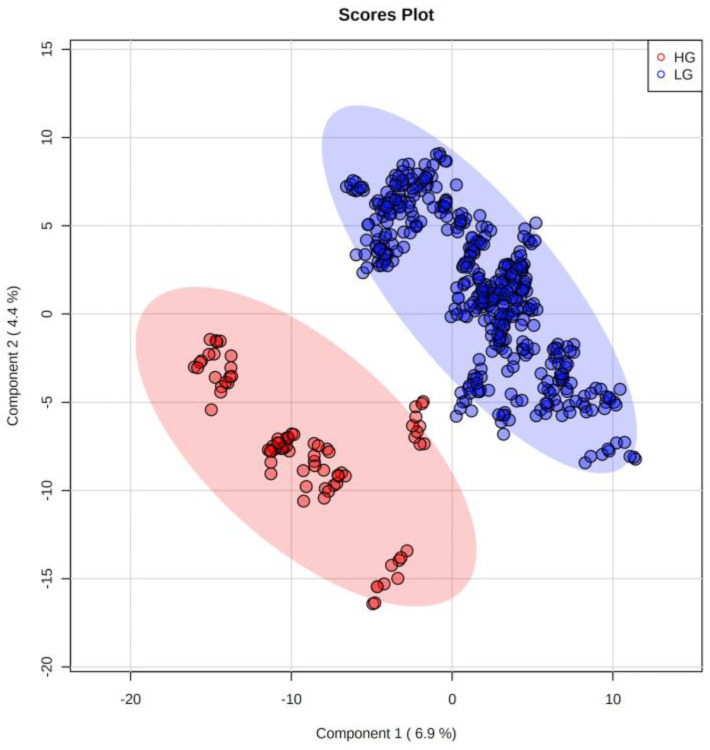
Partial least squares-discriminant analysis of the metabolites revealing the separation between low- and high-grade meningioma cases.

**Figure 2 ijms-24-00394-f002:**
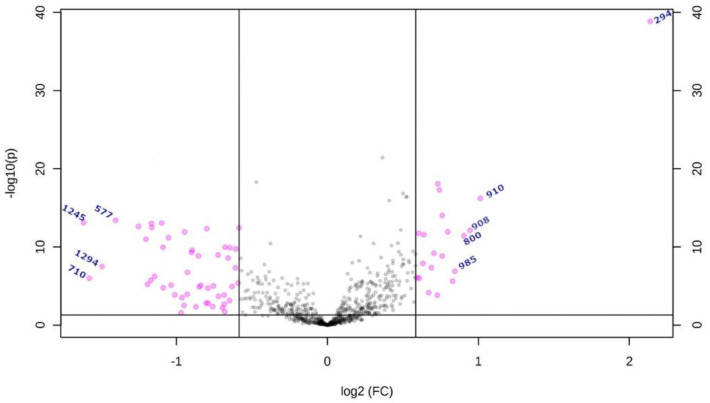
Volcano plot and statistical analysis of the detected metabolites. The *m*/*z* values of metabolites with *p* < 0.05; fold change (FC) ≥ 1.5; Log_2_(FC) ≥ 0.585 were selected (pink dots). The gray dots correspond to *m*/*z* values that does not fit in *p* and FC threshold values.

**Figure 3 ijms-24-00394-f003:**
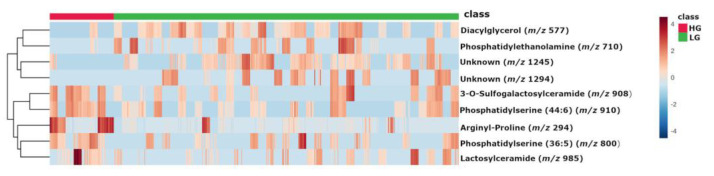
Heatmap analysis of the selected metabolites depending on the meningioma (MGM) group class (represented by the red/green horizontal bar on the top of the graphic). HG, High-grade; LG, Low-grade. The color grade represents the intensity of each biomarker in the patient samples.

**Figure 4 ijms-24-00394-f004:**
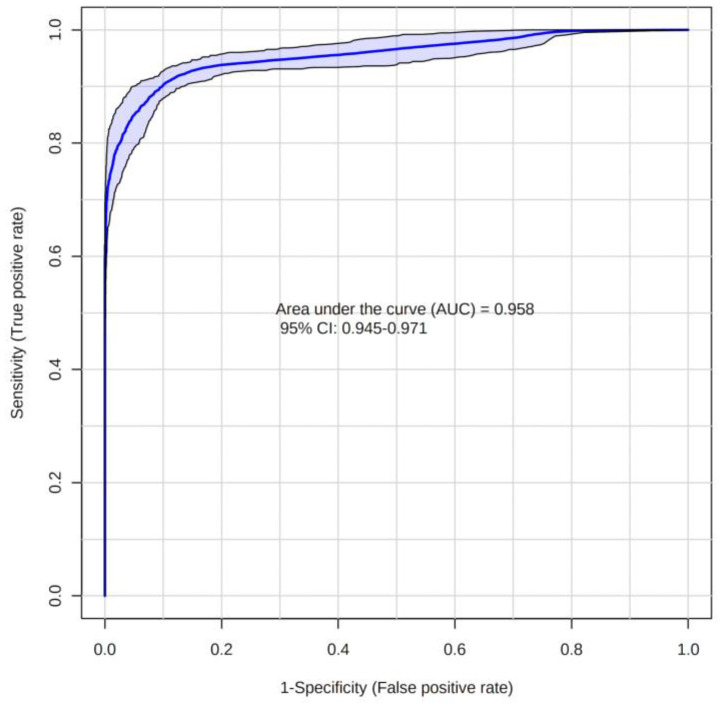
Receiving operating characteristic (ROC) curve analysis to determine the accuracy of the method to designate metabolites that can indicate high-grade meningioma and cancer progression.

**Figure 5 ijms-24-00394-f005:**
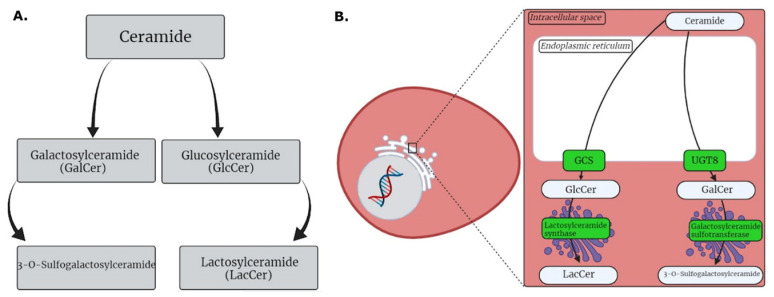
Graphical presentation of sphingolipid metabolism. (**A**) Ceramide glycosylation pathways adapted from MetaboAnalyst 5.0; (**B**) The proposed escape routes of tumor cells (lactosylceramide biosynthesis and sulfatide biosynthesis) adapted from KEGG MODULE Database.

**Table 1 ijms-24-00394-t001:** Low-grade and high-grade meningioma prognostic biomarkers.

	*m*/*z*	Database ID	Metabolite	Formula	Adducts	MS/MS	Log_2_(FC)
High-grade	294	MID85632	Arginyl-Proline	C_11_H_21_N_5_O_3_	[M + Na]^+^	268-254-250-236-266	2.1383
	910	HMDB0116777	PS (44:6) ^#^	C_50_H_86_NO_10_P	[M + NH_4_]^+^	184-104-125-720-495	1.0127
	908	HMDB0012318	3-O-Sulfogalactosylceramide (42:2) ^#^	C_48_H_91_NO_11_S	[M + NH_4_]^+^	494-95-81-604-109-184	0.94786
	800	LMGP03010137	PS (36:5) ^#^	C_42_H_72_NO_10_P	[M + NH_4_]^+^	184-119-368-135-86-437	0.90497
	985	HMDB0011594	LacCer (40:1) ^#^	C_52_H_99_NO_13_	[M + K]^+^	360-338-369-648-437	0.84326
Low-grade	577	MID58656 MID58684	DG (34:1) ^#^	C_37_H_70_O_5_	[M + H-H_2_O]^+^	95-81-57-69-109-339-121-419-135	−1.4045
	1294	-	Unknown	-	-	1275-1262-1243-798	−1.4927
	710	MID40644	PE (32:0) ^#^	C_37_H_74_NO_8_P	[M + NH_4_]^+^	661-655-678-692-674-642-341-454	−1.5768
	1245	-	Unknown	-	-	112-907-184-720-338-625-303	−1.6156

The biomarkers were chosen based on *p* values (*p* < 0.05) and fold change (FC) (FC ≥ 1.5; Log_2_(FC) ≥ 0.585). Abbreviations: *m*/*z*, mass/charge ratio; ID, identifier of the metabolites in the databases researched; MID, METLIN; HMDB, Human Metabolome Database; LMGP, LIPD MAPS; ^#^ (carbon number: double bond number); DG, diacylglycerol; LacCer, lactosylceramide; PE, phosphatidylethanolamine; PS, phosphatidylserine; MS/MS, tandem mass spectrometry.

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
