# Peer review of "Differential Plasma Metabolites between High- and Low-Grade Meningioma Cases"

_ijms, 2022, doi:10.3390/ijms24010394_

Round 1
Reviewer 1 Report
This study is to identify differential blood plasma metabolites between low-and high-grade meningioma patients with the aim to develop a less invasive technique to evaluate grading classification. Overall, the study was well-designed and the findings are convincing. The following are some comments and suggestions for the study:
1. As mentioned by the authors, one of the biggest limitations of this study is the low sample size for high-grade meningioma patients. It would be great if the authors can validate the findings using some publicly available datasets. These could be some microarray or transcriptomic datasets, with the targets corresponding to the metabolites that the authors found in this study.
2. The identified differential metabolites for high-grade meningiomas are relative to low-grade meningiomas based on this study. From the application perspective, wouldn’t it be more beneficial if the study also includes a comparison with the blood plasma from normal individuals for pre-operative grading classification?
Reviewer 2 Report
Purpose of study was to identify meningioma grades prior to surgery but not clear if plasma samples were collected before surgery. Samples collected after surgery may be affected by medications etc. used during surgery.
Define PLSDA before use acronym.
English needs some work.
Overall, interesting paper.
